# Experiential Value, Satisfaction, Brand Love, and Brand Loyalty toward Robot Barista Coffee Shop: The Moderating Effect of Generation

**Young Joong Kim [1], Jung Sook Park [2]** and **Hyeon Mo Jeon [1,***

1 Department of Hotel, Tourism, and Foodservice Management, Dongguk University-Gyeongju, Gyeongju 38066, Korea; yjkim@dongguk.ac.kr
2 Convergence of Lifelong Education, Sangji University, Wonju 26339, Korea; jspark@sangji.ac.kr
* Correspondence: jhm010@dongguk.ac.kr; Tel.: +82-10-6275-4010

**Abstract:** This study applies experiential value, satisfaction, brand love, brand loyalty, and generation to identify consumer behavior toward robot baristas providing new non-face-to-face services during the COVID-19 pandemic. For the analysis, a set of hypotheses was developed and tested based on the data collected from 404 customers who had visited a robot barista coffee shop (RBCS) in South Korea. The results show that playfulness had the most positive effect on satisfaction, followed by service excellence and consumer return on investment (CROI). Satisfaction had a positive effect on brand love and loyalty. This result indicates that playfulness, service excellence, and CROI are important for inducing brand love and brand loyalty of customers toward RBCS. Moreover, generation plays a moderating role between satisfaction and brand love, and between brand and brand loyalty. This research design and the results differ from those of previous studies on experiential value that have focused on human services in the hospitality industry. Consequently, this study contributes to the hospitality literature by applying the experience value theory, which has been mainly applied to research on human services, to non-face-to-face service research, and to identifying its role. Additionally, it makes an important contribution by presenting practical implications for the sustainable management of the food service industry in the COVID-19 era.

**Keywords:** robot barista coffee shop; experiential value; satisfaction; brand love; brand loyalty; generation

## 1. Introduction

COVID-19 has greatly affected all industries economically and brought about many social and cultural changes; the food service industry is no exception [1]. Consumer lifestyles are heavily impacted by concerns about mandatory lockdowns, social distancing, and the uncertainty brought about by the pandemic [2]. Specifically, the pandemic has continuously created massive uncertainty for the food service industry due to decreased consumer demand for food consumption and reduced frequency of dining out [3,4]. In the worst-case scenario, thousands of businesses, whether recently opened or old, may permanently cease activity [5]. Most small businesses are facing the risk of bankruptcy due to COVID-19 and the resulting predicted economic crisis [6]. For example, restaurant sales are declining in South Korea due to the prolonged COVID-19 outbreak; restaurant owner debt increased by an average of 10,000 USD in one year, and more than half of them were contemplating closing their business [7]. For coffee shop businesses in South Korea, COVID-19 is likely to hinder the sales growth of coffee business, and coffee shop operators are working to minimize their losses by focusing on quarantine [8]. Furthermore, at a Starbucks store near Seoul Station, an employee was infected by a customer; the disease quickly spread to other customers, and eventually, the store was closed for a long time [9]. This example of infection caused by face-to-face service between employees and customers emphasizes the need for non-face-to-face services.

Although many small- and medium-sized restaurants fail in the first four years of business, indicating the need for financial strategies to sustain their business beyond five years,

restaurant businesses play an essential role in the national economy [10]. In the context of COVID-19, other authors have also pointed out the importance of survival strategies for the food service sector and restoring consumer confidence [3,6,11,12]. Therefore, it is time to discuss the sustainable management of the food service industry in the COVID-19 era.

The hospitality industry, which has been struggling to survive and recover during the coronavirus disease 2019 (COVID-19) pandemic, is carefully resuming leisure services by gradually instilling contactless technology to deliver food using robots and bringing innovation at a "safe" distance and space [13]. The development of advanced digital technologies, such as artificial intelligence (AI), the Internet of Things (IoT), and smart robots, has also brought a new trend called "food tech" in the food service sector [14]. Food tech, that is, food and technology, is a new industry that has applied information and communications technology (ICT) to the food service industry. Technological innovation is perceived as a key factor for successful transformation in the post-COVID-19 era, but there were already some significant structural changes happening prior to the pandemic [13]. Therefore, during the COVID-19 pandemic, the choices of goods and services changed, with a heightened focus on safety and reliability, depending on the environmental changes, thereby accelerating the commercialization of food tech [15].

The food tech craze is also spreading in the coffee shop business, and services by robot baristas are increasing in coffee shops [16]. Café X in California set up a robot in a space that resembles a fish tank, and a system-operated tablet for customers to place orders while interacting with the machine [17]. Robot baristas by OrionStar in China are replacing human baristas, adding water with a kettle over ground coffee beans, and brewing coffee manually [16]. BARIS CAN, the coffee robot at the Korean robot barista café Lounge X, offers services from receiving orders to serving coffee, and the unmanned Café AI offers opportunities for new cultural experiences [18]. These robot baristas provide complete "contact-free" services suitable for the pandemic. Robots are becoming a typical means of service in the contact-free era, beyond industrial machines [19].

However, even though an increasing number of hospitality companies have adopted service robots in their operations in the last few years, some argue that the effect of this technology is not satisfactory. A study titled "Robot hired" explains that the robot recently hired at a supermarket in Scotland was fired after a week because it could not understand customers [20]. Henn-na Hotel, operated in Nagasaki, Japan, was the first hotel where all staff members were robots, but approximately half of them were fired due to some technical failures [21]. Some hospitality companies in China have also withdrawn robots from the front lines and are planning to reconsider their service robot application strategies [22].

Thus, there is considerable dissatisfaction with robot service, along with their positive aspects in the hospitality industry, which is why studies must be conducted from the perspective of consumer behavior before the supply and diffusion of robot service. Nonetheless, there is insufficient research on robot service experience in the hospitality industry, especially regarding research focusing on customer experience in robot barista services in the coffee consumption market. Therefore, this study focuses on how experiential value in robot barista services in robot barista coffee shop (RBCS) affects customer satisfaction, brand love, and brand loyalty.

Experiential value is customer awareness of the values that they have actually experienced in a service situation [23]. The importance of experiential value is extensively explained in hospitality studies. Previous studies reveal that creating value by improving customer experience indicates an effort to improve relationship qualities (satisfaction and trust) that affect customer behavior and loyalty [24,25]. Experiential value in the hospitality industry has been mostly applied to studies on the use of general human service stores [23–26], but in this study, it is applied to RBCS, a type of contact-free service that is widely used nowadays. This is the biggest difference from previous studies that applied experiential value.

Satisfaction with contact-free service use in the hospitality industry encourages consumers to make repeated purchases of products or services [27], and customer satisfaction

has a positive effect on their intention to recommend the product or service to others [28]. This study also applies brand love as a mediator of customer satisfaction and loyalty in RBCS. Brand love is a component that is completely different from satisfaction, and is a response experienced by some satisfied consumers [29]. Satisfaction is generally conceptualized as a cognitive judgment, but brand love has a much stronger emotional focus [29]. In previous studies, brand satisfaction is referred to as a precursor of brand love, and higher satisfaction leads to brand love that will result in the spread of positive word-of-mouth (WOM) among customers [30]. Brand love is a new component of marketing [30,31]. Competition is rapidly increasing in the food service industry, and it is difficult to retain customers as they tend to pursue diversity [32]. Therefore, an in-depth understanding is needed on the factors that lead to brand love from the perspective of hospitality business [33].

Therefore, based on previous studies [24–26], this study will evaluate how experiential value contributes to the formation of satisfaction, brand love, and loyalty. According to previous studies, there is a difference in awareness and consumer behavior in various products and services among generations [24,34]. Therefore, this study will determine the potential moderating role of generations, assuming that there will be a difference in the relationship between experiential value, satisfaction, brand love, and loyalty in RBCS, depending on the generation. This research design is differentiated from previous studies on experiential value that have focused on human services in the hospitality industry [24–26]. The results will provide important antecedents for predicting consumer behavior in RBCS. Based on the results, this study will provide useful practical suggestions for building long-term relationships with customers for the sustainable management of RBCS during and after the COVID-19 pandemic.

## 2. Literature Review and Hypotheses

### 2.1. Experiential Value

The overall consumption experience is an important aspect of consumer value perception in the service industry [24,35,36]. Holbrook [37] defines consumer values as "interactive and relative preference experience", which emphasizes the transaction between users and the products from which the values are derived [25]. The concept of experiential value refers to the relative preference perceived by customers vis-à-vis the product attributes or service performance they are provided with in service encounters [36]. Experiential value perception is closely related to the interaction between actual use and the remote evaluation of goods and services, and this interaction provides the foundation for experiential value perception [26,38].

However, to fully understand how consumers evaluate products and services, researchers suggest that the multidimensional conceptualization of value perception is necessary because it enables the perception of certain levels of values in relation to one another, rather than individually [24,36]. Mathwick et al. [36,38] examined how the benefits of consumer experience create new experiential value depending on the value of goods and services [23,26]. Mathwick et al. [36] conceptualized and developed measures of experiential value in four dimensions: consumer return on investment (CROI), service excellence, aesthetics, and playfulness. First, CROI refers to the practical aspect of the consumption process, which involves active investment in terms of economic, time, behavioral, and psychological resources traded in return for experience. More specifically, CROI reflects the relative amount of utility obtained by customers in transactions in return for investment in their money, time, and effort [26,36]. Second, service excellence reflects generalized consumer appreciation in service providers, fulfilling their promises through proven expertise and work-related performance [36,39]. Third, aesthetics represent the consonance and unity of physical objects in terms of overall performance [26]. This is further divided into the dimensions of visual appeal and entertainment [26]. Visual appeal indicates the noticeable visual elements of the environment, whereas entertainment is defined in terms of the service environment (or service itself) that excites customers [26].

Finally, playfulness includes escapism and enjoyment [40]. Playfulness is defined as the extent to which consumers perceive a purchasing activity as enjoyable and can break free from everyday needs [38,40].

Consumers have their own expectations and/or previous experience prior to encountering certain products and services, which is considered a part of the experience and affects satisfaction during and after the process of experiencing the products and services [41]. A few studies on hospitality have proven a significant relationship between experiential value and satisfaction. Jin et al. [26] proved that CROI, service excellence, and aesthetics have a significant effect on the satisfaction of full-service restaurant customers. Taylor Jr. et al. [24] conducted a study on customers of pop-up dining, and discovered that exciting escape combining aesthetics and escapism, service excellence, and CROI have a significant effect on satisfaction. In a study on online shopping, Keng et al. [35] analyzed the effect of experiential value (including CROI, aesthetics, service excellence, and playfulness) in shopping malls on satisfaction. Rezaei and Valaei [39] examined consumers who experienced shopping on smartphones, and classified experiential value into four factors (CROI, service excellence, aesthetics, and playfulness) to determine their significant relationship with satisfaction. As such, in previous studies on experiential value, these four factors have been proven to be significant antecedents that improve customer satisfaction. Therefore, the four factors of experiential value in this study are suggested as important predictors of customer satisfaction, even in the RBCS. Based on the results of previous studies, the following hypotheses were established:

**Hypothesis 1 (H1).** *CROI has a significant positive effect on satisfaction.*

**Hypothesis 2 (H2).** *Service excellence has a significant positive effect on satisfaction.*

**Hypothesis 3 (H3).** *Aesthetics has a significant positive effect on satisfaction.*

**Hypothesis 4 (H4).** *Playfulness has a significant positive effect on satisfaction.*

### 2.2. Satisfaction

Satisfaction has been regarded as a key marketing theory and an important purpose of marketing strategies for over 60 years [42]. The key results of marketing activities have changed the initial expenditures and purchase into post-purchase behavior, such as repurchase and brand loyalty; thus, satisfaction has secured a central and dominant position in marketing theories and practices [43]. Oliver [44] explains that satisfaction is "judgment that a product or service feature, or the product or service itself, provided (or is providing) a pleasurable level of consumption related fulfillment" [25]. Satisfaction is a positive response to the outcomes of previous experience, and occurs when a product's performance is as high as the consumer's expectations [45]. Moreover, satisfaction is defined as not only a purchase experience or specific consumption outcome, but also as a type of perception that is accompanied by the evaluation process, whereby consumers compare their anticipated outcome with what they actually obtain [46]. Therefore, satisfaction is regarded as a response of fulfillment used to understand and evaluate consumer experience, indicating a change in attitudes due to the consumption experience [25].

There are two ways to measure satisfaction: the first is based on specific transactions, and the second is based on overall satisfaction [47]. Transaction-specific satisfaction is explained as customer evaluation of the consumption experience and the response regarding specific product transactions, episodes, or service encounters [48]. Overall satisfaction is explained as the overall customer evaluation of a product or service provider [48,49].

Previous studies argue that brand satisfaction is a precursor of brand love, but not all satisfied customers must have a love for the brand [30]. Higher satisfaction leads to brand love, which may promote positive WOM among customers [30]. A few studies on hospitality have proven that customer satisfaction serves as a predictor of brand love and has a positive effect on brand love. Drennan et al. [50] discovered that satisfaction with

wine brands has a positive effect on brand love. Shen et al. [51] argue that brand love is formed among customers who are satisfied after experiencing fine dining restaurants. Cuong [43] discovered that satisfaction with fast food restaurants is a significant antecedent in forming brand love. Based on the results of previous studies, the following hypothesis was established:

**Hypothesis 5 (H5).** *Satisfaction has a significant positive effect on brand love.*

Marketing studies have provided empirical evidence that satisfaction is a powerful indicator of repeated purchases and recommendations of products or services, which are key components of loyalty [49]. Moreover, satisfaction is a prerequisite for brand loyalty, and an increase in satisfaction leads to an increase in brand loyalty [45]. Many studies on hospitality have discovered a positive relationship between customer satisfaction and brand loyalty. Shapoval et al. [52] discovered that loyalty has a positive effect on customer satisfaction with green restaurants. Hidayat et al. [53] found that customer satisfaction with fast food restaurants is an antecedent that increases customer loyalty. Raduzzi and Massey [54] state that the satisfaction of McDonald's customers has a significant positive effect on brand loyalty. Based on the results of previous studies, the following hypothesis was established:

**Hypothesis 6 (H6)**. *Satisfaction has a significant positive effect on brand loyalty.*

*2.3. Brand Love*

Brand love has recently been perceived as a marketing concept in research trends in consumer–brand relationships [31,55]. Brand love is an important outcome for brand managers and plays a strategic role in building sustainable consumer–brand relationships in the long run [55]. Fournier (1998) [56] focuses on the importance of brand love and presents it as a long-term connection between clients and the brand [43]. Brand love is defined as the degree of passionate emotional attachment a satisfied consumer has for a particular trade name [29]. Keh et al. [57] define brand love as an intimate, passionate, and devoted relationship between consumers and the brand, and suggest mutual, purposive, and dynamic attributes as their characteristics [45]. Brand love includes passion for the brand, attachment to the brand, positive evaluation of the brand, positive emotions toward the brand, and declaration of love for the brand [39].

Satisfaction is regarded as the result of each transaction connected to the expectancy–disconfirmation paradigm, but brand love is the result of an unrequired long-term relationship between consumers and the brand [29]. In other words, brand love is an indicator of consumers' emotional responses to a brand [33,58]. Therefore, brand love, which is a stronger emotional variable than brand attitude and preference, becomes a priority for marketers [33,58].

A few studies on hospitality have shown that brand love is a predictor variable of customer loyalty and has a positive effect on loyalty. Liu et al. [59] discovered that brand love for hotels increases revisit intention and positive WOM among millennial consumers. Song et al. [60] discovered that love for a coffee shop brand is an antecedent that increases brand loyalty. Shen et al. [51] argue that brand love formed after experiencing a fine dining restaurant increases recommendations by, as well as the revisit intention of, customers. Cuong [43] discovered that brand love for fast-food restaurants has a positive effect on brand loyalty. Based on the results of previous studies, the following hypothesis was established:

**Hypothesis 7 (H7).** *Brand love has a significant positive effect on brand loyalty.*

*2.4. Brand Loyalty*

Creating loyal customers is the main goal for service marketers, and it occasionally shows the basic components of a company's long-term competitive strategy [26]. Oliver [61] defines brand loyalty as a deeply held commitment to rebuy or patronize the brand

consistently in the future, despite situational influences and marketing efforts with the potential to cause switching behavior [46]. Bowen and Shoemaker [62] explain that loyalty shows the possibility that customers will participate in various purchasing behaviors in the future, including perceived product/service superiority, refund business, social ties, and business recommendations. Chou [63] states that brand loyalty reduces marketing costs and increases relationships between sellers, thereby reducing threats to competitors, arguing that loyal customers are more likely to purchase more products/services and recommend them to others [43,63].

According to Oliver [61], general brand loyalty can be divided into attitudinal and behavioral loyalty, thereby emphasizing the two different aspects of brand loyalty. More specifically, attitudinal loyalty focuses on the psychological commitment to the store or brand, while behavioral loyalty represents the concept of revisit, purchase frequency, and recommendation [26,64]. This study used a behavioral approach to measure brand loyalty.

*2.5. Moderating Effect of Generation*

A generation is an identifiable group that shares similar years of birth and important life events in their life development stages [59]. Gen X refers to those born between 1961 and 1980, whereas Millennials were born between 1981 and 1999. However, researchers have different views of this classification. Millennials are also referred to as Gen Y [65]. Gen X have memories of economic recession and corporate downsizing, and have experienced financial insecurity, family instability, lack of strong traditions, and great diversity [59,66]. Gen X lack the social skills of their parents, but tend to have strong technical capabilities [67]. People of this generation are likely to find wise and quick ways to deal with things, even if they must bend the rules [68]. Gen X are characterized mainly by individualism, self-reliance, and skepticism [69].

Millennials were born in the Information Age, when people began to depend greatly on ICT in everyday life [70]. Therefore, they search for and consume information more actively than the previous generation, and also use ICT more frequently when placing orders or making reservations [59]. Millennials have grown up in a media-saturated environment and have stronger brand consciousness [71]. They focus more on satisfaction and trust in products and services compared with other generations [72]. However, they are often swayed by friends and celebrity endorsements, and tend to switch brands more quickly than other generations [73]. They also show a preference for unique stores and something special that creates a new, memorable experience [59]. Thus, although Millennials eat out more often than other generations, their eating habits and dining preferences still confuse many restaurant managers [73].

On the one hand, Millennials value socially oriented and modern lifestyles, have a strong self-identity, and tend to seek new hedonic experiences [59]. On the other hand, Gen X consumers want to hear more explanations about product features and why these features are necessary [74]. They also have a risk aversion tendency and low risk acceptability [69]. Therefore, there may be a difference between Millennials and Gen X in the desire to experience RBCS, which have become more widely used as a non-face-to-face service during the COVID-19 pandemic, along with the development of Fourth Industrial Revolution technologies.

Taylor Jr. et al. [24] tested the moderating effect of the generation on the relationship between the experience value of pop-up restaurants, relationship quality (satisfaction and trust), and behavioral intention. In particular, relationship quality (satisfaction and trust) has a stronger effect on the future return intention of Millennials than Gen X. Moreover, Taylor Jr. and DiPietro [73] measured the significant moderating effect of generation on the relationship between satisfaction with the restaurant service environment and repeat patronage intention. Brand love for hotels has been proven to be a key variable that affects the revisit and positive WOM of Millennial consumers [59]. However, while previous studies have provided interesting results, they have not examined the moderating role of brand love [33,58], that is, the indicator of consumers' emotional response to the brand.

Therefore, this study suggests generation as a moderating variable between the experiential value of RBCS, satisfaction, brand love, and loyalty, and evaluates the moderating effect of a generation to reduce the knowledge gap with previous studies. Therefore, the following hypotheses are established:

**Hypothesis 8 (H8).** *The influence of (a) CROI, (b) service excellence, (c) aesthetics, and (d) playfulness on satisfaction differ according to generation.*

**Hypothesis 9 (H9).** *The influence of satisfaction on brand love differs according to generation.*

**Hypothesis 10 (H10).** *The influence of satisfaction on brand loyalty differs according to generation.*

**Hypothesis 11 (H11).** *The influence of brands love on brand loyalty differs according to generation.*

All the hypotheses are in the theoretical model, as depicted in Figure 1.

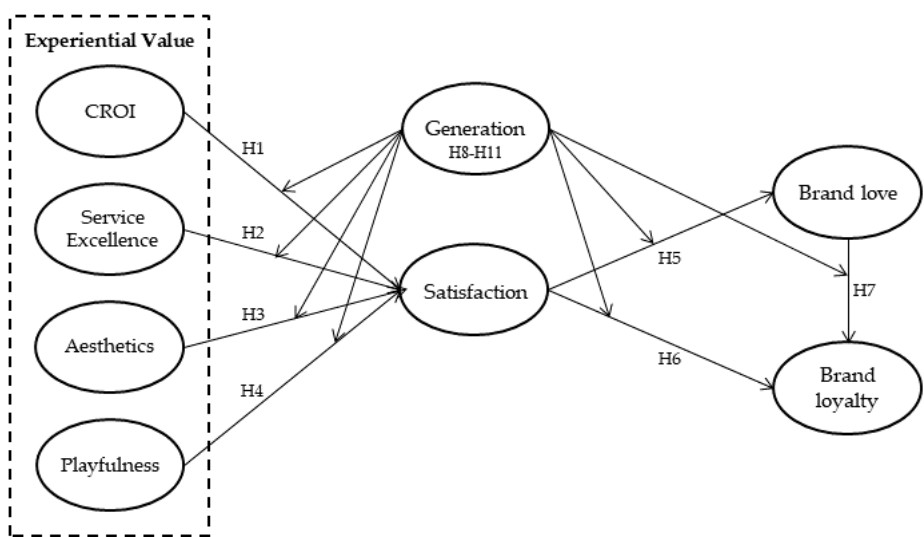

**Figure 1.** Research model.

### 3. Methodology

*3.1. Research Instrument*

This study primarily develops survey items based on previous literature, and revises them to be more suitable for RBCS. Second, the first questionnaire was developed, and experts comprising three food service professors and three RBCSs owners were requested to provide opinions and reviews on the contents and items of the questionnaire. Before the actual survey, a pilot test was conducted to determine whether the participants could fully understand the items. Items that were difficult to accurately measure were deleted using exploratory factor analysis, after which the items were revised to more appropriate expressions for the final questionnaire.

The questionnaire comprised seven factors, namely: perceived CROI, service excellence, aesthetics, playfulness, satisfaction, brand love, and loyalty. Four items on CROI were cited from studies by Kim and Stepchenkova [23], Taylor et al. [24], and Tsai and Wang [75], and three items on service excellence were cited from studies by Rezaei and Valaei [39], Taylor et al. [24], and Tsai and Wang [75]. Four items each on aesthetics and playfulness were cited from studies by Kim and Stepchenkova [23] and Tsai and Wang [75]. Four items on satisfaction were cited from studies by Song et al. [76] and Wu [77], and four items on brand love were cited from studies by Cuong [43] and Liu et al. [59]. Three items on loyalty were cited from studies by Sung and Jeon [16] and Wu [77]. All items were rated on a five-point Likert scale from "strongly disagree" to "strongly agree".

### 3.2. Sampling and Data Collection

The participants for the sampling were men and women living in Korea that were born between 1961 and 1999, and who used a RBCS in the last three months. Café AI, Lounge'X, Dalkomm b;eat, and coffee methods were selected as the target brands to examine, which were found to be the most commonly used RBCS among coffee consumers as a result of the pilot test. Gen X and Millennials are the customers who mainly use RBCS; thus, baby boomers were excluded from the participants. The customers of these four coffee shops are representative of all RBCS customers.

Data were collected online through Entrust from 6 to 27 January 2021. Entrust is a global online research company with offices in Korea and Hong Kong, and is a reliable survey agency with a panel of over 560,000 people. The subjects were asked whether they had visited the four coffee shop brands and purchased a product made by a robot barista in the last three months. Participants were asked if they had visited one of the four RBSCS selected and if they had ordered something prepared by a robot barista in the last three months. The purpose and intent of this study were explained to those who had experience, and the survey was conducted after obtaining their consent. The survey was provided to all respondents in the same order, and 404 samples were ultimately used in the empirical analysis. As the response rate was 11.2%, this dataset was subjected to a non-response bias analysis using wave analysis. Responses collected in the first seven days were classified as "initial responses", and those collected in the last seven days were categorized as "latter responses". These two groups were both used to conduct an independent samples *t*-test, and the difference between the two was not significant. This indicates that there was no nonresponse bias within the dataset.

The demographic characteristics of the sample were examined and specified as follows: 45.8% of the respondents were male and 54.2% were female; 19.3% of the respondents were between 20 and 29, 32.7% were between 30 and 39, 24.3% were between 40 and 49, and 23.7% were between 50 and 59 years old. The majority of the respondents (72.5%) had obtained at least a university degree; 18.6% had an average monthly income of USD 3000–3999, 15.1% between USD 4000 and 4999, 15.1% between USD 2000 and 2999, and 12.9% between USD 5000 and 5999. Lastly, 38.2% were office workers, 18.2% were professionals, and 17.2% were in sales and services.

### 3.3. Analytical Methods

This study used SPSS 22.0 and AMOS 22.0 as analytical tools. First, the demographic characteristics of the respondents were analyzed using SPSS 22.0. Data analysis for hypothesis testing was conducted using the two-step approach (measurement model and structural model assessment) by Anderson and Gerbing [78]. Confirmatory factor analysis (CFA) was conducted to test the fit of the measurement model and to assess convergent and discriminant validity. Thereafter, structural equation modeling (SEM) was conducted to test the relationships among the seven constructs suggested in the conceptual model and the moderating effect of generation.

## 4. Results

### 4.1. Measurement Model

The fit of the measurement model was assessed using CFA. To estimate the fit of the model, seven common model fit measures were used: $\chi^2/df$ (<3) GFI (>0.90), RMSEA (<0.08), RMR (<0.08), NFI (>0.9), IFI (>0.9), and CFI (>0.9) [79]. Table 1 provides the CFA results after eliminating one item each from CROI, playfulness, and satisfaction, which reduced the model fit based on squared multiple correlation. The measurement model had a good fit with the data collected ($\chi^2$ = 565.453, df = 259, $\chi^2/df$ = 2.183, RMR = 0.034, GFI = 0.908, NFI = 0.911, IFI = 0.950, CFI = 0.949, and RMSEA = 0.054) (see Table 1). The fit of the measurement model was assessed based on reliability, convergent validity, and discriminant validity. Reliability was examined based on the composite construct reliability (CCR). Table 2 shows the adequate CCR, with all values exceeding 0.7 [79]. The average

variance extracted (AVE) values of all variables were greater than the suggested threshold of 0.5, thereby implying the convergent validity of the scale [79].

**Table 1.** Measurement model assessment.

| Variables and Item | SL [a] | CCR [b] | AVE [c] |
|---|---|---|---|
| **Customer return on investment (CROI)** ($\alpha$ = 0.704) | | | |
| Using the RBCS has a good economic value | 0.684 | 0.776 | 0.535 |
| Using the RBCS is convenient | 0.639 | | |
| Using the RBCS is a more efficient way to manage my time than using the general coffee shop | 0.676 | | |
| **Service excellence (SE)** ($\alpha$ = 0.729) | | | |
| Robot baristas have professional coffee menu manufacturing skills | 0.580 | 0.768 | 0.528 |
| The service at the RBCS makes me feel more special and valued than the service at general coffee shops | 0.700 | | |
| When I think of robot barista, I think of excellence | 0.773 | | |
| **Aesthetics (AT)** ($\alpha$ = 0.759) | | | |
| The robot barista's appearance is very impressive | 0.652 | 0.829 | 0.548 |
| The interior environment of the RBCS is aesthetically appealing | 0.694 | | |
| Using the RBCS makes me feel good | 0.711 | | |
| The atmosphere of the RBCS is wonderful | 0.589 | | |
| **Playfulness (PF)** ($\alpha$ = 0.792) | | | |
| Using the RBCS makes me feel like being in another world | 0.786 | 0.872 | 0.684 |
| The using experience at the RBCS was truly a joy | 0.700 | | |
| Compared with other things, the experience at the RBCS was truly enjoyable | 0.760 | | |
| **Satisfaction (SF)** ($\alpha$ = 0.721) | | | |
| The RBCS goes beyond my expectations | 0.653 | 0.817 | 0.598 |
| I think I did the right thing when I experienced the RBCS | 0.693 | | |
| I am satisfied with the robot barista staff | 0.706 | | |
| **Brand love (BL)** ($\alpha$ = 0.915) | | | |
| This RBCS brand is totally awesome | 0.859 | 0.897 | 0.685 |
| I love this RBCS brand | 0.881 | | |
| I am passionate about this RBCS brand | 0.876 | | |
| I am very attached to this RBCS brand | 0.800 | | |
| **Brand loyalty (BT)** ($\alpha$ = 0.836) | | | |
| I will continue to visit the RBCS brand in the future | 0.766 | 0.870 | 0.690 |
| I am highly likely to recommend the RBCS brand to friends | 0.815 | | |
| I will spread positive word-of-mouth about the RBCS brand | 0.797 | | |

Note: [a] Standard loading, [b] Composite construct reliability, [c] Average variance extracted.

**Table 2.** Correlations of analysis between the variables.

| Variable | 1 | 2 | 3 | 4 | 5 | 6 | 7 |
|---|---|---|---|---|---|---|---|
| 1. CROI | 0.731 | | | | | | |
| 2. SE | 0.389 | 0.727 | | | | | |
| 3. AP | 0.335 | 0.409 | 0.740 | | | | |
| 4. PF | 0.356 | 0.412 | 0.473 | 0.827 | | | |
| 5. SF | 0.398 | 0.466 | 0.448 | 0.532 | 0.773 | | |
| 6. BL | 0.247 | 0.431 | 0.147 | 0.250 | 0.261 | 0.828 | |
| 7. BT | 0.399 | 0.444 | 0.407 | 0.499 | 0.559 | 0.413 | 0.831 |
| Mean | 3.868 | 3.688 | 3.964 | 3.876 | 3.887 | 3.082 | 3.780 |
| S.D. | 0.657 | 0.725 | 0.616 | 0.653 | 0.616 | 1.000 | 0.760 |

Note: Diagonal elements show the square root of AVE. Below the diagonal is the corresponding correlation coefficient. All correlation coefficients are significant at the 0.001 level.

To examine the discriminant validity of the variables that have been proven to have convergent validity, the square root of the AVE of each latent variable was compared with the relevant correlation coefficient among the latent variables. Table 2 shows that the square root of the AVE of each latent variable was greater than the correlation coefficient, thereby showing an adequate discriminant validity [80].

### 4.2. Structural Model

SEM was conducted using the AMOS 22.0 statistical package. The fit of the structural model describing the relationships among constructs was assessed to test the hypotheses established through the SEM path coefficients. The model fit indices were $\chi^2$ = 501.282, df = 204, $p$ = 0.000, $\chi^2$/df = 2.457, RMR = 0.041, GFI = 0.903, NFI = 0.911, IFI = 0.945, CFI = 0.944, and RMSEA = 0.060, thus meeting the standard assessment criteria. The results of each hypothesis test describing the causal relationship between any pair of constructs are presented in Table 3. H1 was supported because CROI positively and significantly influenced satisfaction ($\beta$ = 0.362, t = 2.861, $p$ = 0.004). H2 was supported because service excellence positively and significantly influenced satisfaction ($\beta$ = 0.623, t = 3.594, $p$ = 0.000). H3 was rejected because aesthetics did not significantly influence satisfaction ($\beta$ = −0.574, t = −1.636, $p$ = 0.102). H4 was supported because playfulness positively and significantly influenced satisfaction ($\beta$ = 0.649, t = 2.267, $p$ = 0.023). H5 was supported because satisfaction positively and significantly influenced brand love ($\beta$ = 0.683, t = 11.371, $p$ = 0.000). H6 was supported because satisfaction positively and significantly influenced brand loyalty ($\beta$ = 0.799, t = 11.060, $p$ = 0.000). Lastly, H7 was supported because brand love positively and significantly influenced brand loyalty ($\beta$ = 0.190, t = 3.386, $p$ = 0.000).

**Table 3.** Results of the structural model analysis.

| Hypothesis | | β | t-Value | *p*-Value | Decision |
|---|---|---|---|---|---|
| H1 | CROI → SF | 0.362 | 2.861 ** | 0.004 | supported |
| H2 | SE → SF | 0.623 | 3.594 ** | 0.000 | supported |
| H3 | AP → SF | −0.574 | −1.636 | 0.102 | rejected |
| H4 | PF → SF | 0.649 | 2.267 * | 0.023 | supported |
| H5 | SF → BL | 0.683 | 11.371 ** | 0.000 | supported |
| H6 | SF → BT | 0.799 | 11.060 ** | 0.000 | supported |
| H7 | BL → BT | 0.190 | 3.836 ** | 0.000 | supported |

Note: * $p$ < 0.05, ** $p$ < 0.01.

### 4.3. Moderating Effect of Generation

To empirically determine the moderating role of generation, this study used a multiple-group analysis. To evaluate the differential effects of the moderator, the chi-square difference between the unconstrained and constrained models was assessed in terms of the difference in the degrees of freedom [81] (see Table 4). First, the moderating effect of generation on the relationship between experiential value and satisfaction was evaluated (H8a, H8b, H8c, and H8d). The chi-square difference between the unconstrained and constrained models was not significant at the 0.05 level. Generation did not play a moderating role in the relationships between (1) CROI and satisfaction ($\Delta\chi^2$ = 0.888 < $\chi^2_{0.05}$(1) = 3.84, df = 1), (2) service excellence and satisfaction ($\Delta\chi^2$ = 0.283 < $\chi^2_{0.05}$(1) = 3.84, df = 1), (3) aesthetics and satisfaction ($\Delta\chi^2$ = 0.405 < $\chi^2_{0.05}$(1) = 3.84, df = 1), and (4) playfulness and satisfaction ($\Delta\chi^2$ = 0.326 < $\chi^2_{0.05}$(1) = 3.84, df = 1). Thus, H8 was not supported. Second, the moderating effect of generation on the relationship between satisfaction and brand love was evaluated (H9). The chi-square difference between the unconstrained and constrained models was significant at a 0.05 level. Generation plays a moderating role in the relationship between satisfaction and brand love ($\Delta\chi^2$ = 4.000 < $\chi^2_{0.05}$(1) = 3.84, df = 1). Thus, H9 was supported. Third, the moderating effect of generation on the relationship between satisfaction and brand loyalty was evaluated (H10). The chi-square difference between the unconstrained and

constrained models was not significant at the 0.05 level. Generation played a moderating role in the relationship between satisfaction and brand loyalty ($\Delta\chi^2 = 1.031 < \chi^2_{0.05}(1) = 3.84$, df = 1). Thus, H10 was not supported. Fourth, the moderating effect of generation on the relationship between brand love and brand loyalty was evaluated (H11). The chi-square difference between the unconstrained and constrained models was significant at a 0.05 level. Generation played a moderating role in the relationship between brand love and brand loyalty ($\Delta\chi^2 = 12.016 < \chi^2_{0.05}(1) = 3.84$, df = 1). Thus, H11 was supported.

**Table 4.** Results of the moderating effects.

| Hypothesis | Millennials (*n* = 210): β (t-Value) | Gen X (*n* = 194): β (t-Value) | Unconstrained Model $\chi^2$ (df = 400) | Constrained Model $\chi^2$ (df = 401) | $\Delta\chi^2$ (df = 1) |
|---|---|---|---|---|---|
| H8a | 0.325 (1.991 *) | 0.123 (0.889) | | 871.153 | 0.888 |
| H8b | 0.526 (2.338 *) | 0.613 (2.379 *) | | 870.308 | 0.283 |
| H8c | −0.063 (−0.254) | −0.405 (−1.640) | | 870.670 | 0.405 |
| H8d | 0.271 (1.284) | 0.692 (3.083 **) | 870.265 | 870.591 | 0.326 |
| H9 | 0.572 (7.200 **) | 0.791 (8.384 **) | | 874.265 | 4.000 ** |
| H10 | 0.816 (8.802 **) | 0.573 (6.360 **) | | 871.296 | 1.031 |
| H11 | 0.116 (1.908) | 0.435 (5.589 **) | | 882.281 | 12.016 ** |

Note: * $p < 0.05$, ** $p < 0.01$.

## 5. Discussion

The results of the data analysis confirm that the satisfaction of the RBCS was affected by playfulness, service excellence, and CROI. In particular, playfulness had the greatest influence on satisfaction. The customers were very surprised and delighted by how the robot barista made and served orders from the menu, and the level of satisfaction with the menu and service provided was high. Additionally, the convenience and promptness of ordering and service provision improved time management, and the reasonable price increased the economic value; thus, user satisfaction increased. This result is consistent with previous hospitality studies [24,26] related to experiential value. However, aesthetics did not have a significant effect on satisfaction. This result is in contrast with previous hospitality studies [24,26] related to experiential value. This is thought to be because customers perceive the aesthetic elements of RBCS as lower than the aesthetic elements provided by the restaurant.

Moreover, it was confirmed that satisfaction increased brand love, which is consistent with existing hospitality studies [43,50,51]. It can be interpreted that customers who are satisfied with RBCS are more likely to maintain a long-term relationship with them. Additionally, it was confirmed that satisfaction increased brand loyalty, which is consistent with previous hospitality studies [52–54]. This suggests that customers satisfied with RBCS are more likely to revisit, recommend, and use word-of-mouth. Additionally, brand love increased brand loyalty as well, consistent with previous hospitality studies [43,51,59,60]. Customers who wanted to maintain a long-term relationship with RBCS indicated that they were more likely to revisit, recommend, and use word-of-mouth. The result of the moderating effect of generation shows that there is no intergenerational difference in the relationship between experiential value and satisfaction, as in previous study on human services [24]. However, there is no intergenerational difference in the relationship between satisfaction and loyalty, thereby showing a different pattern from that of previous study [24]. Moreover, there is an intergenerational difference in the relationship between satisfaction and brand love, and between brand love and brand loyalty. This result indicates that Gen X has a stronger emotional attachment to maintaining a long-term relationship with RBCS than Millennials.

### 5.1. Theoretical Implications

One of the most important theoretical implications is related to the theory of experiential value used in this study. Experiential value has been used in many studies as a cognitive evaluation of the values directly experienced by customers in service encounters. However, there are limitations in that they have failed to examine customer satisfaction with the new service experience in the non-face-to-face service environment applying Fourth Industrial Revolution technologies. To overcome these limitations, this study applies the theory of experiential value to robot service representing non-face-to-face services, which is different from previous studies on human service experience [24,25,52]. Moreover, this study intends to overcome the limitations of satisfaction as a cognitive result by applying brand love, which is an indicator of customers' emotional responses to the brand [33,58] and represents long-term relationships with customers [29]. This design differs from previous studies [16,22] on robot service in the hospitality industry.

This study tested the relationship among factors and identified the most important antecedents for the satisfaction of customers who experienced RBCS. The results confirmed that the experiential value theory can be applied to the robot service context. However, unlike the human service context, aesthetics was not related to satisfaction in the robot service context. Consequently, we confirmed that only playfulness, service excellence, and CROI, among the sub-factors of experience value in the robot service context, were related to customer satisfaction. This has a theoretical contribution as the results are different from the experiential value studies [24,26] that have mainly been applied to human services. Moreover, satisfaction does not only have a direct relationship but also an indirect relationship with brand loyalty through brand love. Consistent with the hospitality sector literature [43,50,51], it was confirmed that the relationship among satisfaction, brand love, and brand loyalty was a major result of consumer behavior, even in the context of robot service.

We also analyzed the moderating effect between generations, assuming that the use of robot service, excluding human service, may vary depending on the generation. The effect of satisfaction on brand love and the effect of brand love on brand loyalty were greater among Gen X than among Millennials. This result offers a theoretical contribution by confirming that the role of brand love in the emotional response of customers to RBCS and that the maintenance of long-term relationships varies by generation.

Finally, this study applies experiential value, satisfaction, brand love, brand loyalty, and generation to identify consumer behavior toward robot baristas providing new non-face-to-face services during the COVID-19 pandemic. It also determines the structural relations among the variables and verifies the fit of the model. The research design and results contribute to studies on robot service in the hospitality industry. Therefore, the theoretical framework proposed and verified in this study will be the foundation for research on non-face-to-face services and customer experiences preparing for the post-COVID-19 era in the food service business. Furthermore, this study is useful for sustainable food service management, especially at the point where social discussions are needed due to the digital transformation accelerated by COVID-19.

### 5.2. Practical Implications

It is necessary to focus on playfulness, service excellence, and CROI to improve customer satisfaction and brand love for RBCS from a practical perspective. Coffee consumers in Korea are fascinated by robots making and serving coffee while using RBCS and enjoy the experience in itself. With the robot doing something that has been hitherto done by a human barista, customers feel as if they are in a futuristic world. Thus, developers and coffee shop owners must collaborate to analyze the coffee making motions of human baristas with big data and apply these to AI robot baristas, thereby using more advanced and elaborate service technology on the robots. For example, robot baristas with AI algorithms must be able to identify the consumption patterns of specific customers, such as preferred coffee products, amount of purchase, and purchase frequency, by scanning their eyes or

fingerprints. With this data, robot baristas must also be able to recommend drinks on the menu to customers. They must also provide detailed explanations about products through chatbots, such as bean type, roast level, taste, and flavor. These innovative service features of robot baristas will provide convenience to customers and improve their satisfaction, thereby increasing their love for the brand. Although robot baristas can still offer only limited coffee products, their innovative service features will allow customers to perceive the expertise of robot baristas and experience new values at RBCS.

By providing seamlessly connected services that enable quick orders without any hassle to increase CROI, it will be possible to reduce the time required for ordering and serving, thereby enhancing convenience and efficiency. Moreover, as the coffee shop can save personnel expenses in the long run by hiring robot baristas, this cost reduction can be reflected in the prices on the menu, so that more reasonably priced products can be offered. Consequently, customers will receive price benefits, which will further improve their satisfaction and brand love. Meanwhile, the aesthetics and atmosphere of RBCS do not satisfy customers as much as human service coffee shops. Robot baristas that are commercialized today are shaped like robotic arms. Thus, by developing humanoid robot baristas to provide the aforementioned innovative services, this dissatisfaction can be resolved to a certain extent. Moreover, more aesthetic elements must be considered when creating an indoor environment, including interior decorations.

Previous studies argue that not all satisfied customers have brand love, and that higher satisfaction leads to brand love [30]. Therefore, for long-term brand–consumer relationships, RBCS owners must establish more intensive marketing strategies for Gen X, whose satisfaction leads to brand love relatively better. For Gen X with risk aversion tendency and low risk acceptability, there must be tighter quality management of coffee products in RBCS. Moreover, it is necessary to promote convenience in using the ordering and payment system for Gen X consumers who are not familiar with using ICT compared to Millennials.

Even after the COVID-19 pandemic ends, customers who have become accustomed to non-face-to-face robot barista services will continue to use RBCS. As non-face-to-face services are rapidly increasing due to COVID-19, robot service should be established as a sustainable business model for food service businesses, including coffee shops, in the future.

## 6. Limitations and Future Research

This study has the following limitations and sets the direction for future research. First, data are collected only in Korea, and thus the results cannot be generalized. As the development of Fourth Industrial Revolution technologies may vary among nations, it might be inappropriate to apply the results of this study to other nations. Second, the survey was conducted online to follow social distancing guidelines due to the COVID-19 pandemic. As online surveys may generate selection bias [82], future research must employ various types of data collection methods, such as on-the-spot inspection, to reduce biases and increase response rates. By conducting a comparative analysis of experiential values between RBCS and human service coffee shops, more in-depth discussions can be conducted on strategic measures for sustainable management.

**Author Contributions:** Y.J.K. and H.M.J. conceived and designed the experiments; H.M.J. performed the experiments and analyzed the data; Y.J.K., J.S.P. and H.M.J. wrote the paper. All authors have read and agreed to the published version of the manuscript.

**Funding:** This work was supported by the Dongguk University Research Fund of 2020.

**Institutional Review Board Statement:** Not applicable.

**Informed Consent Statement:** Not applicable.

**Data Availability Statement:** The data presented in this study are available upon request from the corresponding author.

**Conflicts of Interest:** The authors declare no conflict of interest.

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
