# Peer review of "Experiential Value, Satisfaction, Brand Love, and Brand Loyalty toward Robot Barista Coffee Shop: The Moderating Effect of Generation"

_sustainability, doi:10.3390/su132112029_

Round 1

Reviewer 1 Report

The authors have presented a very interesting manuscript on some contemporary marketing concepts. Although the paper is comprehensive and of high quality, I am not convinced that it is suitable for this journal. I have read again and again the aim & scope and the subject areas of Sustainability and I cannot see how the content of the manuscript is related to them. Maybe the authors should clarify this in their introduction. This is the reason why my suggestion is to "reconsider after revision". This revision is about connecting the manuscript with any of the subject areas of the journal. 

Reviewer 2 Report

- Generally, the paper has some originality.

- The scope of the research is very interesting and valuable.

- The paper gives a very clear idea of the originality and the value provided in the research method and results extracted.

Note:

- Make sure you demonstrate clearly the theoretical contribution of your paper and that is evident in your abstract and in particular in your findings. 

- Discussion of the results needs to improvements, comparing results with those of other authors/papers/studies. 

Round 2

Reviewer 1 Report

The authors have managed to connect their work with the aim and scope of the journal. Therefore, I suggest that their paper is published.